# Identifying and evaluating barriers for the implementation of machine learning in the intensive care unit

Ellie D'Hondt [1✉], Thomas J. Ashby [1✉], Imen Chakroun[1✉], Thomas Koninckx[2] & Roel Wuyts [1✉]

## Abstract

**Background** Despite apparent promise and the availability of numerous examples in the literature, machine learning models are rarely used in practice in ICU units. This mismatch suggests that there are poorly understood barriers preventing uptake, which we aim to identify.

**Methods** We begin with a qualitative study with 29 interviews of 40 Intensive Care Unit-, hospital- and MedTech company staff members. As a follow-up to the study, we attempt to quantify some of the technical issues raised. To perform experiments we selected two models based on criteria such as medical relevance. Using these models we measure the loss of performance in predictive models due to drift over time, change of available patient features, scarceness of data, and deploying a model in a different context to the one it was built in.

**Results** The qualitative study confirms our assumptions on the potential of AI-driven analytics for patient care, as well as showing the prevalence and type of technical blocking factors that are responsible for its slow uptake. The experiments confirm that each of these issues can cause important loss of predictive model performance, depending on the model and the issue.

**Conclusions** Based on the qualitative study and quantitative experiments we conclude that more research on practical solutions to enable AI-driven innovation in Intensive Care Units is needed. Furthermore, the general poor situation with respect to public, usable implementations of predictive models would appear to limit the possibilities for both the scientific repeatability of the underlying research and the transfer of this research into practice.

**Plain language summary**

It is helpful for clinicians to be able to predict what will happen to a patient in an Intensive Care Unit (ICU); accurate computer-based predictive systems could help to avoid serious illness. However, most ICUs currently make little or no use of them. Here, we try to understand why, so that barriers to their introduction can be overcome. We interview medical experts, who agree that prediction systems should be feasible. They also identify practical technical problems with using them. We investigate these issues by running experiments on example predictive systems where we change what data is used to train the system and what data it is asked to make predictions on. The experiments show that the identified issues cause problems and are worthy of further attention. This work should help to enable the use of computer-based predictive systems in ICUs.

[1] Exascience Life Lab, imec, Leuven, Belgium. [2] Independent Consultant, Leuven, Belgium. ✉email: ellie.dhondt@imec.be; Tom.Ashby@imec.be; Imen.Chakroun@imec.be; Roel.Wuyts@imec.be

There is a wide-spread belief in the high potential of data-driven innovation in healthcare[1–3]. Several examples of successful AI implementation in single healthcare centers, commonly using retrospective datasets, show great promise[4]. Although AI-driven healthcare is an active area of research, we see very few of the proposed techniques driving patient care in practice.

This paper presents a qualitative and quantitative investigation into the issues around slow uptake of artificial intelligence (AI) in healthcare. We are most interested in uncovering technical gaps in current research. We focus on the use of predictive machine learning (ML) in the intensive care unit (ICU), because there is rich ICU data monitoring and a large number of ML models proposed in the literature, but a dearth of use in practice. This mismatch indicates possible hidden issues that need to be dealt with to unlock the apparent potential. This is in contrast with the use of ML on medical images, an area similarly rich in data where progress appears to be smoother[5].

Our investigation relies on a fieldwork Voice-of-the-Customer (VoC) study where we interview hospital and MedTech company staff about current practises and needs in ICUs, and the role AI plays therein. This exercise aims to unearth issues related to the building and deployment of ML models. We then proceed to investigate these issues with experiments that simulate how an ML model might get built and deployed in various healthcare scenarios. These scenarios are informed by the VoC interviews with ICU stakeholders as well as the literature. The aim of the experiments is to verify and put some quantitative scope on the technical blocking factors rather than to propose any particular solution. VoC questions were phrased according to the more generic AI, while experiments focused on predictive ML approaches, a subset of AI. To perform experiments, we use selected two models based on criteria such as medical relevance and reproducibility. As a side effect, our search into models satisfying these criteria unearthed the general poor situation with respect to public, usable implementations of predictive models.

Related work pertains to qualitative as well as in quantitative studies. We could not find publications, which couple qualitative research with a subsequent quantitative analysis. Little qualitative research into AI in healthcare exists. Two studies were executed around the same time as ours and published recently. One uses an online survey to poll healthcare practitioners in image-driven medical disciplines[5]. Another interviews 18 general practitioners on their views on AI[6]. Optimism and concern emerged from these works. None discuss overcoming blocking factors or possible innovations. To our knowledge, no qualitative study was carried out in ICUs.

Some quantitative studies discuss blocking factors and possible solutions to them[7]. Some are practical (e.g., the requirement for a coherent digital infrastructure) and others societal (e.g., privacy concerns). Another issue is that data across hospitals can be very different because observational variables such as protocol and locations are hospital-specific. This reality is confirmed by Johnson et al.[8], where in-hospital mortality models trained on publicly accessible data exhibit decreased performance when applied to external hospital data. Transfer learning is a collection of techniques to transfer a model from a training context to a different inference context. Several approaches have been applied to medical data[9,10]. Like us, the authors of these articles hypothesized that external models implemented in a straightforward manner fail to reflect individual variations across hospitals.

Drift over time is similarly considered as a reason for a drop in model performance. Drift was analyzed earlier for the same dataset that we use[11], for predictive models that are, by their own admission, not particularly challenging in that their performance saturates with very little training data. Subbaswamy et al.[12] propose an approach using a single dataset to identify drift problems for arbitrary models; such sensitivity analysis may form part of the toolbox for dealing with drift. In this article, we analyze actual recorded drift rather than potential drift.

Harutyunyan et al.[13] formulate the challenge of increasing the use of ML in healthcare as a benchmarking exercise. However, the set-up is optimized for having a level playing field between ML algorithms rather than solving the issues that we highlight here. In addition the medical relevance of their prediction targets could be better.

Other work attempts to mix data from different hospitals[14,15], considering only one type of model and a limited number of the issues we address in this paper. Further quantitative analysis of their results is hampered by a lack of open-source models.

The results from our study can be summarized as follows. From the VoC we conclude that AI solutions are generally considered to be useful in ICU care and that blocking factors for their slow uptake are mostly technical. Blocking factors that emerge are: the need for more data, the discrepancy between clean research data and dirty real-world data, and algorithms not being sufficiently generalizable nor reliable. We explore these blocking factors in a number of concrete scenarios for building models under changing ICU contexts. The scenarios we explore are: model performance changes over time; evolving a model with extra features that were not there when the model was originally built; model performance dependency on the volume of data; and model performance dependency on patient demographics, by deploying a model in a different hospital than the one in which it was built. For each of these scenarios we show that model performance can be affected. We speculate that the negative impacts on performance can be mitigated and are as such not ultimately blocking for data-driven healthcare.

## Methods

**Voice-of-the-Customer study**. A Voice-of-the-Customer (VoC) study is a type of qualitative study that is considered a proven research methodology to determine the needs, expectations and pain points of a certain user group. Our approach to the VoC process is based on living labs methods relevant for the idea stage of innovative products or services. These methods drive participatory co-design via the methodology of interviews[16,17]. The interview is used as a vehicle for user assumption testing, i.e., checking whether assumptions held by the interviewers about innovation needs are actually true. Among these, leap-of-faith assumption are the most risky assumptions: if proven false, this breaks the innovation idea entirely and sends you back to the starting line. Personas help in grouping contextual user insights and further drive user experience design[18].

The interview structure follows the Double Diamond design process model[19]. Here, the divergent and convergent stages of the design process (the diamond shapes that give the approach its name) are the current state as is and the future state as could be respectively. We start why as the crucial question, leading to questions on what, only to be translated afterwards into a technological how. This generic structure is shaped into a fixed set of questions, which allow us to assess where respondents fall on the conservative-innovative spectrum. By proposing the innovation only midway through the interview, we avoid biasing answers to the questions on the current state and needs towards the assumptions we want to test.

For this ICU VoC study the interview follows a double diamond consisting of parts: introduction, current practices (team, workflow, infrastructure, data), unmet needs and wants (what is and is not working well, future vision, role of data analytics), innovation pitch, attitude towards the proposed solutions, drivers and barriers

(blocking factors for analytics and possible mitigations), and closing. The full interview can be found in Supplementary Note 1. Its structure was reviewed by two experts on user-centric design who were not connected to this specific study. Innovations were introduced by way of mock-ups shown to the interviewees. The mock-ups presented a data-driven patient cockpit. These were refined as interviews progressed. Here, patient data is divided into clinical chapters (haemodynamics, cardiovascular, renal and so on). This single-point user interface presents data in the same way as clinical staff surveys a patient, rather than as directed by the available equipment. Data-driven insights are visualized in the relevant clinical chapter in the patient cockpit interface. The interviews were conducted by a data innovator and a consultant with a business track record in MedTech. One interview takes about 1–2 h to conduct and its subsequent analysis about 1 day. The study was part of an innovation track at imec, a non-profit precompetitive research center in Belgium. Written informed consent was obtained from all participants, by agreeing to participate in the study. Institutional Review Board approval was not required because the study was a voluntary survey among healthcare and MedTech professionals not related to any specific health information and all data was handled anonymously. No vulnerable individuals participated in the survey. Participants were informed that the data would be collated anonymously for analysis and dissemination.

Five user assumptions are tested. The first is that there is rich data in ICUs. Assumptions 2 through 4 are: Data is not optimally accessible to support clinical care personnel in their workflow; Data analytics and AI are rarely used in actual patient care; Analytics has the potential to be useful for patient care (in general as well as for specific diagnoses). Finally, the fifth assumption is that barriers for using data analytics and AI in the ICU are largely technical. Assumptions are implicit in the interview structure rather than polled explicitly. In the context of Assumption 4 we poll interviewees on which care trajectories could be most impacted by data innovation (diagnoses and the patients that receive them). The leap-of-faith assumptions are Assumptions 4 and 5 and are the main focus of this paper.

*Interview demographics.* The VoC study was carried out with 17 hospitals and 7 MedTech companies in the period of May-December 2020. The focus was on the ICU environment, including units for neonatal (NICU) and pediatric (PICU) intensive care.

An overview of the stakeholder statistics is given in Table 1. ICUs involved were of various sizes: 5 had under 20 beds, 5 between 20 and 30, 2 between 30 and 40, and 3 had over 60 beds. For 2 further institutions, the interviewees were not connected to a particular ICU. The 7 MedTech companies provided insights into the landscape for IT support in hospitals and their ICUs. Four companies provide software solutions in ICUs, 1 is an ICU device manufacturer, 1 works on ICU and hospital-wide interoperability, and 1 provides remote monitoring for implantable cardiac devices.

We spoke with a total of 40 interviewees in 29 structured interviews. With 3 stakeholders we carried out more than one interview (resp. 8, 4 and 1 additional interviews) for refinement and/or because they were particularly interested. These complementary interviews were not counted in the interview statistics. Table 2 summarizes the demographics of people we interviewed. Hospital staff generally filled in senior roles in their organization (2 C-level, 4 heads of IT, 7 heads of ICU, 3 head nurses). All had some clinical research activities, with 3 having the bulk of their workload in academic research. MedTech staff interviewees had varied roles: 1 project manager, 3 technical profiles, 1 CEO, 3 sales managers, 1 director market & offering, and 1 senior data scientist. We did not ask interviewees' ages nor ethnicity to protect their privacy. Interviewees were predominantly white.

| Table 1 VoC participant breakdown. | | |
| --- | --- | --- |
| | **Institutions** | **Interviewees** |
| | **n = 24 (%)** | **n = 40 (%)** |
| Type | | |
| Hospital | 17 (71) | 30 (75) |
| Company | 7 (29) | 10 (25) |
| Within hospital: | | |
| ICU | 9 (38) | 13 (32) |
| ICU and IT | 4 (17) | 5 (12) |
| IT | 2 (8) | 9 (22) |
| NICU | 1 (4) | 2 (5) |
| PICU | 1 (4) | 1 (3) |
| Within company: | | |
| HealthIT | 4 (17) | 5 (13) |
| Medical devices | 1 (4) | 2 (5) |
| Interoperability | 1 (4) | 2 (5) |
| Remote monitoring | 1 (4) | 1 (3) |
| Location: | | |
| Belgium | 13 (54) | 27 (67) |
| U.S.A. | 5 (21) | 6 (15) |
| Multiple countries | 4 (17) | 5 (12) |
| U.K. | 1 (4) | 1 (3) |
| The Netherlands | 1 (4) | 1 (3) |

Table 3 summarizes interview statistics. The VoC study was executed during the COVID-19 pandemic. This had an impact on the availability of respondees and hence the timings of interviews. In Belgium, where most of the interviews took place, the peak of the first wave was early April and of the second wave early November. We did not contact anybody during these periods. Of the 61 stakeholders we reached out to outside of these periods, 29 resulted in an interview, 23 were unresponsive, 6 were not available because of other priorities and 3 had to cancel interviews due to COVID-19 priorities. Interviews in October focused on nursing staff, which we wanted to include directly in our study. No interview questions targeted the COVID-19 situation directly. We wanted to obtain an overall view on the potential of data innovation in ICUs, and did not want our answers to be biased with reference to the specific date of the interview. Nevertheless, some responses from ICU staff could have been changed by these external circumstances. We briefly come back to interviewee bias in the analysis of responses related to Assumption 4.

**From VoC to experiments.** The default scenario to the building and deployment of machine learning models is to train local and immediately deploy local. This is a realistic scenario given sufficient data and sufficient expertize and resources to perform the model building. Under these circumstances, a model with reasonable predictive performance can be constructed. The achievable performance in this scenario is usually a ceiling for any other deployment scenarios one could consider.

There is a mismatch between this basic scenario and complications that occur in practice. These complications, and the loss of predictive performance that they entail, are a barrier to wider adoption of ML models. In the quantitative part of this study we attempt to quantify to what extent technical blocking factors impact the performance of analytics solutions. We focus on problems that emerged from the VoC: interoperability issues, working with real-world dirty patient data instead of curated data, the overall lack of (access to) data and the fact that models fail to generalize between the contexts in which they are used. We investigate these problems in four concrete scenarios. We simulate each of these scenarios by building models on subsets of the patient dataset and testing on a different subset,then comparing model performance changes across subsets. This is detailed further in section Experiments.

**Table 2 VoC interviewee demographics.**

| Interviewees | $n = 40$ (%) |
|---|---|
| Sex | |
| Male | 29 (72) |
| Female | 11 (28) |
| Primary role | |
| Physician | 17 (42) |
| MedTech expert | 10 (25) |
| IT expert | 9 (23) |
| Nurse | 4 (10) |
| Seniority | |
| Management | 20 (52) |
| Other | 19 (48) |

**Table 3 VoC interview details.**

| Interviews | $n$ | % |
|---|---|---|
| Reachout | | of $n = 61$ reached out: |
| Interviews | 29 | 47 |
| No response | 23 | 38 |
| N/A | 6 | 10 |
| Canceled | 3 | 5 |
| Month of interview (in 2020) | | of $n = 29$ interviews: |
| May | 1 | 3 |
| June | 8 | 28 |
| July | 1 | 3 |
| August | 2 | 7 |
| September | 12 | 42 |
| October | 3 | 10 |
| December | 2 | 7 |

The first scenario is that of drift of the model over time within a hospital. This occurs because clinical procedures, staff and the patient population change over time within the same hospital. This will eventually degrade the performance of all models built on historical data, even if from the same institution. This can be viewed as part of the dirty data or failure to generalize problem. The second scenario considers acquiring extra diagnostic machines and being able to use the output of those machines to improve predictions (a.k.a. change of features). This will occur whenever a hospital expands or replaces its equipment. While new information per-patient does not make the pre-existing model worse, not using it can constitute an important missing out on possible improvements. This is part of the interoperability problem (models not being interoperable with different feature sets). The third scenario is the scarceness of data for modeling uncommon or rare conditions. If a condition is complex enough that it requires a large amount of data to build a useful predictive model, and it is also a sufficiently rare condition, it is often not possible to build a highly performant or even a usable model from data available at one hospital, even if it is a large one. This is part of the lack of data issue. Fourth and final scenario is where we move a model from one hospital to a different one with a different set of patients, i.e., a change of population. This is a desired approach for any hospital lacking data, resources or expertize to develop sufficiently performant models themselves (the default scenario we see in academic literature). It is also a future vision that emerged from the qualitative study, one where cooperating hospitals divide the labor of building models and share models with each other after they are built. A different hospital implies different clinical procedures, staff and patients, and these changes can degrade the performance of a model, even if it was recently built. This is part of the failure to generalize problem.

A further note on the relevancy of these technical blocking factors: it may seem that the use of a fully developed commercial ML solution would insulate hospitals from the technical issues that we have outlined. However, this solution merely transfers the responsibility for dealing with those issues to the company that is providing the predictive model. This is confirmed by our VoC study; one MedTech company commented that its core business is affected by the lack of generalizability. They mention the long development times needed to adapt their predictive analytics to a specific ICU unit to arrive at actionable performance. Several hospitals reported non-technical difficulties with importing Analytics-as-a-Service solutions, e.g., stating that at least 4 or 5 such service agreements are in the pipeline, but blocked by data protection issues. Hence understanding these issues is of interest to whoever ends up providing or using a model, be that a research group, a hospital or a commercial model provider.

**Selecting models**. To simulate model building scenarios in a realistic way, the first step is to identify a predictive model to use and adapt to each scenario. Based on the qualitative study we arrive at a number of criteria for selecting models for our experiments from the scientific literature. In this section we describe these criteria, the systematic search for predictive models satisfying those criteria, and the final choice of models. The narrative underlines the practical difficulties surrounding the identification of suitable externally developed models for hospitals that do not have the resources to develop their own from scratch.

*Clinical relevance*. When choosing the models to work with in the quantification exercise, we considered the clinical relevance of models. First, is the model sufficiently clinically actionable and medical-outcome orientated? Second, is the condition that the model is predicting sufficiently challenging? Here we considered both the degree to which the models are used for important and uncommon/rare conditions, and a substantial complexity of models so that the technical issues identified in the VoC will have an impact.

The VoC analysis, in particular Assumption 4, allows us to make a first separation of ICU conditions into those that are clinically relevant and those that are not. In this way, we ruled out mortality prediction, length of ICU stay, and phenotype prediction[13]. Indeed these conditions are less relevant to medical aspects of care, as they lack actionability and/or are not sufficiently challenging (e.g., they can be trained with little data due to relatively shallow insight or the condition being common). As such these models are less likely to suffer from the issues we are trying to quantify. This reasoning is partly described by Nestor et al.[11] for length-of-stay and mortality prediction, and partly based on the results of the VoC.

*Reproducibility*. One learning scenario that emerged clearly from the VoC is that of reusing a model locally that was learned in another context. To do experiments for this scenario, it is essential to have a model that is available, can be retrained on different datasets and that is of sufficient initial quality. This leads to a number of criteria that models need to satisfy on top of clinical relevance, which we refer to as reproducibility. The need for retraining means in practice that the source code to build the model and relevant input data need to be available and usable. Concretely, we say that a ML model satisfies reproducibility if the model's source code is publicly available, if it operates on a high-quality dataset that is publicly available and if it is backed by a peer-reviewed publication, to give some guarantee of correctness of approach. Furthermore, the source code has to come with a compatible and explicitly declared license in the repository to ensure that use and/or modification of the code is legally acceptable, and we have to be able to run it after only investing a reasonable amount of effort. A lack of license was considered not legally

acceptable. Unfortunately many public repositories have no license, which means that they cannot be reused, e.g., for reproducibility of results, due to the legal ambiguity around what a lack of license means for users and/or publishers of code. Dealing with different national jurisdictions also complicates this analysis.

*Model search and selection.* We conducted the search for reproducible and clinically relevant model repositories by using search engines, both directly on GitHub, the largest store of publicly available repositories that we could find, and in a general internet search using Google. Our search focused on the two main publicly available ICU datasets that existed at the time of our experiments (circa 2020 and 2021): the single-center ~38 K patient MIMIC-III[20,21], and the multicentre ~200 K patient eICU[22,23] critical care database. We used PhysioNet, an online catalog of physiological and clinical data and related open-source software, as an additional search term. More specifically, we searched for 7 search phrases. On Github: `mimic-iii physionet`; `physionet`; `e-ICU`; `eicu`. On Google: `mimic3 github` `icu prediction`; `physionet`.

In all cases, we processed the first 10 pages of results (or all of them if there were fewer), which equates to up to 100 items per search phrase. This approach was inspired by the ideas behind systematic review.

We give here a brief overview of the search effort. Given that one of the main effects that we are trying to understand is the failure of models to generalize from one context to another, models that work on multi-hospital datasets are a natural avenue of investigation. Hence we focused first on eICU. However, we did not find any eICU models that matched both clinical relevance and reproducibility. We then looked for models on the single-center MIMIC-III dataset, on the grounds that it is also large, high quality, easily available and well known, while being slightly older, thus more analysis work would have been published on it. Here, we could find models that matched our criteria. Note that this search was not directed towards a list of specific conditions, but for any condition that we thought would have some clinical relevance.

In total, we found 27 repositories that were of sufficient quality to be interesting, but had no associated scientific article; 18 repositories with an article that were either making a model that we considered not clinically relevant, a model that was not predictive of a clinical condition (e.g., inferring the presence of a clinical condition post facto given all the data of a stay); and 8 more interesting repositories that are discussed in more detail in Supplementary Note 2. From these 8, only one was clinically relevant and reproducible. This is the model we chose; its topic is readmission prediction and it is further described by Pakbin et al.[24] and in section Readmission prediction. As we did not find another model satisfying all criteria, we decided to develop one in-house. We chose AKI as a condition to model because of its clinical relevance and the willingness to share clinical domain knowledge on this condition from one of the stakeholders we interviewed. To keep the experimental work consistent, we used MIMIC-III for our AKI model, and made our model publicly available.

One other source of models that we did consider using was the 2019 PhysioNet challenge: early detection of sepsis in the ICU. While there are numerous public repositories available related to this dataset accompanied by publications, the data appears to be no longer publicly available. In addition, the dataset is not that different from MIMIC-III in that the data available to the challenge participants came from only two different hospitals, and thus the number of different contexts was limited and not a big gain over taking the data from a single institution.

In general, we found few scientific publications come with any public repository and dataset at all. Conversely, various public repositories lack an associated peer reviewed article, making it very hard to assess their scientific worth without a huge investment of time and effort. Of the repositories that are public and backed with an article and public dataset, a fair amount are for models that are of little interest to ICU practitioners based on our VoC results. In the limited number of cases of interesting models, almost all of them suffer from licensing problems, making them legally problematic, and some would seem to suffer poor implementation quality, meaning that they would require a potentially large investment to get to run. While some of these problems should be easy to fix (e.g., licensing), others, such as the general lack of available models and the problems around clinical relevancy are likely to require more discussion and action within the community of people working in this area.

## Model details

*Acute kidney injury prediction.* AKI is a complex medical phenomenon that is uncommon, but which has an important impact on both short and long-term mortality when it does happen[25]. Kidney injury emerged in our VoC study as one of the more interesting target diagnoses for AI-driven innovation, and was suggested as a candidate model to adopt in our patient cockpit mock-ups by one stakeholder. Hence AKI prediction is clinically relevant, challenging to model, and rare enough that total volume of training data can be an issue. Given that we wanted to run experiments on more than one model and there were no other options available among public models, it seemed a reasonable choice. For this model we drop the reproducibility requirement of peer-review as is less important in the context of these experiments; the aim is not to get an actionable clinical AKI model, but rather to understand how models behave when put into different scenarios. We have made the model publicly available. Various publications on AKI predictors have been made that either have no public repository, or where the repository does not match our criteria[26–29].

The AKI (acute kidney injury) model we define predicts the probability of developing AKI or not within 7 days of admission to an ICU. AKI is a rapid decline in renal function associated with long hospital stay, elevated healthcare charges and high mortality risk especially in ICUs[25]. AKI affects 5% to 7% of all hospitalizations and causes 10 billion dollars of additional healthcare-related expenditures per year through per-hospitalization excess costs of $7933[30]. No interventions to improve outcomes of established AKI have yet been developed, so prevention and early diagnosis are key[31]. Ability to predict the onset of AKI could also be of help as a first step in the discovery and assessment of new therapies.

We construct a deep-learning model that predicts AKI in the following way. First, we use the Kidney Disease/Improving Global Outcomes (KDIGO) criteria[32] to define whether patients have developed one of the AKI three stages of increasing severity (Stage 1, Stage 2, Stage 3). Next, we developed a detection model for AKI stage on the basis of 83 input features referring to routinely collected clinical parameters. Features include demographics data, vital signs measured at the bedside (heart rate, arterial blood pressure, respiration rate, etc.), laboratory test results (blood urea nitrogen, hemoglobin, white blood count, etc.), average of urine output (when available), the minimum value of estimated glomerular filtration rate and creatinine. We also included comorbidities such as congestive heart failure, hypertension, diabetes, etc. We constructed a set of multilayer perceptrons and used hyperparameter tuning to find the best architecture (a 15-layer deep-learning network) for predicting AKI stage from these parameters. We followed a 80% training–20% testing partition of data with five-fold cross validation.

*Readmission prediction.* Similarly to AKI, ICU readmission is uncommon, and is associated with worse medical outcomes

compared to patients that are not readmitted. Although readmission has a similar level of occurrence to AKI, it is a much more general phenomenon (there can be many reasons for readmission), and may therefore be difficult to model as accurately.

The VoC underlines the relevance of (re)admission prediction for planning purposes and, more importantly, for timely movement from another hospital ward to the ICU to improve clinical outcomes. Discharge from an ICU means that a patient will be leaving an environment with close monitoring. This entails a higher risk that deterioration in their condition may go unnoticed. Readmission to an ICU is associated with increased risk of adverse events, longer hospital stays and worse mortality outcomes[33]. Worldwide, ~6 to 7% of patients are readmitted to an ICU within 72 h. Accurate prediction of readmission gives doctors an actionable tool at a critical decision point (whether or not to admit a patient to / discharge a patient from the ICU), which can help them better plan the use of ICU resources as a whole and avoid unnecessary risks for the patient; if a patient is predicted to be readmitted, then they are clearly a candidate for keeping in the ICU for further monitoring.

The readmission model that we use was forked from an open GitHub repository (see section on Code Availability). The code required some modifications to be adapted to our local systems, but was relatively easy to get working. The model is actually a collection of models to predict readmission within different time windows (we refer to the original paper for the explanation of these). Throughout the rest of the paper we refer to this collection as one model, but we present the results for all of the constituent models.

*Cohorts.* Cohort selection for AKI proceeded by extraction from MIMIC-III, excluding patients with AKI upon admission in the ICU or chronic kidney diseases and patients under 18. During the prepossessing step, we omit entries with missing values necessary for computing AKI stage via the KDIGO guidelines using creatinine and urine output. This resulted in a dataset of 42,152 ICU stays, with 16,837 labeled as non-AKI patients, 7558 as Stage-1 AKI, 13,535 as Stage-2 AKI, and 4321 as Stage-3 AKI.

Cohort selection from the dataset for the readmission model is described in their publication and repository. Similarly, train and test split and preparation etc. is described. The total data size after their preparation steps is 53,329 distinct ICU stays. To understand the impact of splits, we establish baselines for both models. These are the models trained using the full cohort and a random split for train and test (80%–20%).

**Experiments**. In order to quantify the problems for the four different learning scenarios identified earlier, we performed ML experiments using the selected AKI and readmission models. These experiments consisted of training the model on one subset of MIMIC-III and testing on another subset, to understand how the prediction performance of models is affected. To capture drift we subset by EHR system, which is a proxy for date. Change of features is simulated by in- or excluding urine output information in the patient record. Scarceness of data is represented by random subsampling of available training data. Finally, a change of population is arrived at by sub-setting by ethnicity. This is a proxy for possible changes in patient populations across different hospitals. The performance metric is area under receiver operator curve (AUROC).

*Scenario 1: Drift.* To understand how drift affects models, we compare the baseline model against a model that is trained on time-span A and tested on time-span B, where A and B are non-overlapping. Simulating drift on the basis of a time partition in

**Table 4 Data partitions and percentages.**

| Data partition | Ethnicity | | Drift | |
|---|---|---|---|---|
| | White | Other | Before cut-off | After cut-off |
| **Whole dataset** | **71.6** | **28.4** | **50.0** | **38.6** |
| Ethnicity: White | | | 49.5 | 39.4 |
| Ethnicity: Other | | | 51.3 | 36.7 |
| Drift: Before cut-off | 70.8 | 29.1 | | |
| Drift: After cut-off | 73.0 | 27.0 | | |

Data partitions in MIMIC-III and the associated percentages of other categories in these data partitions. The two partitions of interest of the whole dataset are shown in bold (split by ethnicity or split by date cut-off for drift). The other numbers show how the percentages for those main partitions change when applying another partitioning first (given in the left-most column).

MIMIC-III is not possible due to de-identification methods (i.e., dates were shifted into the future by a random offset). Hence we looked for a proxy for time. Sometime circa 2008, the institution collecting the MIMIC-III data moved from the CareVue to the Metavision EMR systems. The EMR system used is visible in the data, and can be used to delineate time spans. The split of data is about 50% pre-2008 and 40% post-2008, with about 10% of patients with entries in both the CareVue and Metavision systems during a cross-over phase (cfr. Table 4). Note that the percentages for drift do not add up to 100% because there is 11.4% of patients who appear in both the CareVue and Meta-Vision systems and who are not used for training nor testing to keep the time cut-off as clean as possible. We ignore cross-over patients in our analysis below.

To assess how drift affects selected models, we train the model on a subset of the data pre-2008 and test it on the subset post-2008 and ignore patients that appear in both systems. The sizes of the extracted subset are 24347 for the pre-2008 (CareVue) and 17806 for the post-2008 (Metavision). The same data split on the readmission data led to 26682 pre-2008 and 20627 post-2008 distinct ICU stays. Again, we compare the baseline model against the performance of a model that is trained on pre-2008 data only, and then tested on post-2008 data only.

*Scenario 2: Change of features.* For the AKI model we simulate this scenario by simulating a hospital that evolves its infrastructure by adding a device for urine measurements to its ICU setup. This feature is an addition to the medical parameters previously collected. The hospital wants to use this new feature to enhance its previously developed AKI predictor. The straightforward way to deal with this is to train the new model from scratch. In this case, the old model is deprecated and a new cohort is initiated where the urine feature is added. While this is easy to set up in principle, in practise it may require a long time until enough data is collected to build a new predictor with better predictive performance than the original predictor, given that the average number of patients per year admitted to ICUs ranges between 100s (small hospitals) up 1000s (large hospitals). To quantify this, we use subsets of the data to train the model using the extra features and compare this to the final performance on all the available data of the model without the extra features.

To further explore this issue, we also quantify the benefit of using basic transfer learning on the old model to adjust to the availability of new features (the urine measurements), and compare the resulting model's performance to that of the model described above using urine features retrained from scratch. This gives an idea of how much headroom there is for improvement using more advanced forms of transfer learning. Our simple transfer learning is implemented by freezing part of the neural network that

implements the AKI model predictor, while allowing other layers to adapt to the novel data including urine features.

An experiment for scenario 2 was not performed for readmission prediction. As it is a general rather than disease-specific model, there is no obvious candidate for adding a device that would specifically record a feature known to be linked to the probability of readmission. This would make the addition of a feature rather arbitrary.

*Scenario 3: Scarceness of data.* For this scenario, we train the model on a randomly selected sub-sample of the original input cohort of a given size and report the model performance on the remainder of the data. We plot how the ML performance evolves across various dataset sizes. This shows how much gain in performance is made with each step of additional data.

*Scenario 4: Change of population.* In this scenario, we simulate the change of patient demographics that could exist between different hospitals by splitting MIMIC-III data by ethnicity. The aim is to simulate the drop in performance for the scenario where one hospital imports an ML model developed elsewhere. We use a model trained on one subset of the data (White) to predict outcomes on other subsets (Black and Latinx). The baseline model shown for comparison includes all three ethnicity labels in both training and test sets, with White making up the majority in the dataset.

The sizes of the extracted subsets for the AKI model are 30,618 for White, 3155 for Black and 1440 for Latinx cohorts. The same data split on the readmission data led to 38,184 White (inc. all subgroups), 5070 Black (inc. all subgroups), and 1803 Latinx (inc. all subgroups) distinct ICU stays.

*Confounding of data splits.* When testing performance changes that come with context changes in the data, there is potential for confounding. That is, the categories used for the splits could be correlated, and thus it is incorrect to say that a context-induced change in prediction performance is specifically due to one split or another. However, the aim of the experiments is not to make any strong claims about the impact of any one particular type of split, but rather to show that context changes cause performance loss. Despite this, it gives extra insight if we quantify the potential for confounding. This applies to Scenarios 1 (drift) and 4 (change of population); for scenario 3 we sub-sample the parent dataset at random, and so there is little scope for systematic confounding with other splits. In scenario 2, there is no demographic split: the same population is used, with or without urine features.

We quantify the splits used for training the models compared to the remainder of the data (some or all of which would be used for testing) in Table 4. The table shows that the data splits that we use, namely ethnicity and drift, are only very weakly correlated. As such, any difference in the models is highly unlikely to be due to any change in characteristics resulting from the other category.

**Reporting summary**. Further information on research design is available in the Nature Portfolio Reporting Summary linked to this article.

## Results

**Voice-of-the-Customer study**. This section analyzes the responses to test Assumptions 1–5 for the VoC study on data innovation in ICUs. The interview allows multiple answers to a single question. As such the numbers below do not always add up to the number of stakeholders interviewed.

All 17 ICU units interviewed have similar care workflows, with shifts, handover protocols and touring as common elements. This includes the distribution of typical tasks to personas (senior vs assistant doctors/ nurses). Patient monitoring equipment is also comparable across institutions, even if specific vendors differ. All patients have a bedside monitor showing physiological parameters, alongside a variety of pumps (medication and nutrition). We see that Assumption 1 holds: all hospitals consulted offer rich patient data. However, the way in which this data is stored after patient discharge varies widely. Eight institutions store bulk patient monitoring data (time series frequencies may vary). Five of them do not store data at all, but instead work with discharge forms. For 4 hospitals, the data storage procedure is unknown. Automatic storage is the norm, but 1 of the institutions manually copies data into a complementary Excel file.

Assumption 2, the suboptimal accessibility of data, is widely supported by the VoC study. Two complementary IT systems exist, historically offered by different vendors: The hospital-wide Electronic Medical Record (EMR) system, and the ICU-specific patient data monitoring system (PDMS). Only 1 hospital has an integrated EMR-PDMS, 8 hospitals has separate systems, 5 hospitals has an EMR and no PDMS, and 3 unknown. Data is typically transferred from PDMS to EMR by PDF export/import. That assumption 2 is a real issue is reflected in the future needs cited, with 1 interviewee reporting coupling of EMR-PDMS, 6 interviewees integration of data and 2 dealing with information overload. Alarm fatigue and the need for smarter alerting is mentioned twice; every new piece of technology adds a new alarm[34]. Overall, the way in which data is presented to clinical staff is driven by vendor islands, and not by the clinical view on the patient.

Looking into current practices we see confirmation of Assumption 3: AI-driven solutions to support patient care in ICUs are rare. In fact, none of the ICU units we consulted integrates an AI component into systems or workflow. Two stakeholders mention a (haemodynamics and EEG) device with embedded analytics, one of which is lying around gathering dust, because nobody knows how to use it [sic]. Five clinical stakeholders developed research prototypes, used sporadically in actual care. One unit developed a dashboard to integrate infection treatment data. This dashboard, while known, is not used in nearby hospitals. We do see a widespread use of rule-based systems (12 stakeholders), from fairly standard Sequential Organ Failure Assessment (SOFA) scores up to multi-feature smart alerting (for sepsis and Acute Respiratory Distress Syndrome (ARDS) at 2 stakeholders, and as a product from 1 MedTech stakeholder). One MedTech vendor offers a discharge-readmission software product. We do not include benchmarking tools, which evaluate patient data retrospectively to support hospital and government policy and do not affect patient care directly.

This brings us to the fourth assumption: that AI-driven patient care has potential. Of the 40 interviewees, only 1 did not believe this to be the case, trusting on rule-based systems instead. All other interviewees are convinced that analytics will lead to innovation in patient care to some extent. When polled about their future vision (before proposing our own ideas on data-driven innovations), 10 mentioned AI (3 of which predictive analytics), 1 decision support, 3 support for personalized care, and 3 integration.

We checked which ICU-specific use cases interviewees consider promising targets for AI. A basic widespread need for a 24/7 patient cockpit emerged, an interface which integrates and presents patient data by clinical chapter, just as doctors read out a patient. This idea matured into a mock-up visualization as the VoC study progressed. In the interviews with mock-up, 10 interviewees suggest a patient interface before we showed the mock-up, and 10 confirm its usefulness after we showed it. Specific diagnoses mentioned are: smart trend analysis (5), sepsis (10), kidney failure (4, with 2 adding

this not being very useful), deterioration (3), extubation/ARDS (3), intracranial pressure (2), delirium (2), and outlier detection (2). One person counter-suggested mortality prediction due to its limited usefulness in the ICU.

Mortality prediction, length of stay and discharge-readmission predictive analytics are common topics in the academic literature on health analytics. They are useful for capacity planning purposes and retrospective benchmarking, but less relevant for daily care. This is confirmed by one MedTech company, which markets a discharge-readmission solution. However, two senior intensivists states that (re)admission prediction is relevant for cross-ward patient care trajectories. Timely (re-)admission to ICU environments is crucial for fast-deteriorating pathologies such as sepsis. The differing opinions on predictions of kidney failure stem from the fact that it cannot yet be treated. That said, early detection has an impact on patient comfort and planning of care.

Finally, let us check Assumption 5: are barriers for uptake of AI in the ICU largely technical? Technical blocking factors given by interviewees are: the need for more data (9), interoperability issues (1), privacy and data protection (7), and the reality of dirty real patient data as opposed to curated datasets used for research (7). Non-technical factors mentioned were lack of funding for digitalization (5) and a culture of doctor-knows-best (7). One respondent states that both data and [analytics] algorithms got a lot of babysitting in research publications. In terms of the algorithms we see the following responses: they are not actionable (7), not generalizable (from the setting were they were developed to the care context were you want to use them, (9), not reliable (i.e., model performance is too low, 5) or not explainable (1). One respondent mentions the lack of a killer app, i.e., an app, which is so necessary or desirable for clinical care that it is a core driver to innovation and its uptake inevitable.

There are indications of bias in our interview sample. When setting up interviews, we simply stated the topic to be data innovation in the ICU. There is some risk that those interviewed are self-selected as a sample with positive bias towards this topic. On the other hand, the VoC occurred in COVID-19 times causing extremely busy ICUs. It is unlikely that unresponsiveness was always due to disbelief or disinterest in health data innovation. Two hospitals were involved before the COVID-19 crisis hit in Europe. We asked them in May, respectively September 2020, how the situation impacted their views. The first responded that, aside from an acute and severe lack of resources, the potential of and issues with data analytics in ICUs remain much the same. The second gave insights in the novel patient monitoring needs of the typical COVID-19 patient, and the interplay with sepsis and delirium.

In conclusion, all five user assumptions hold within the VoC's sample set. This includes the leap-of-faith assumptions: AI solutions can be useful in patient care and blocking factors are mostly technical. If AI-driven models are actionable, accurate and compatible with the existing clinical workflow, interviewees go from willing to enthusiastic about their use. There is legitimate concern about the decrease of model performance from a controlled research setting to analytics in the wild [8]. As put by one senior physician: You need to make [... people] see that it works at a practical level.

**Experiments**. The baseline AKI model demonstrates an overall high AUROC of 0.86. The baseline readmission sub-models obtain an AUROC that varies from 0.71 to 0.84.

*Scenario 1: Drift*. The results of Table 5 show the impact in terms of AUROC performance for the AKI model. Drift causes a drop in performance of 0.16 AUROC, which is a big effect.

**Table 5 AKI model: impact of drift and ethnicity.**

| Experiment | AUROC performance |
|---|---|
| Baseline | 0.86 |
| Drift | 0.70 |
| White-Black | 0.78 |
| White-Latinx | 0.79 |

The results for Readmission in Table 6 show that drift always has a negative impact on the performance of the model, which varies from moderate (0.03) to small (0.01).

*Scenario 2: Change of features*. In Fig. 1, we quantify how much data has to be collected to achieve better predictive performance in the new model that uses the additional urine features. This gives an idea of the length of time when there is missed benefit due to inferior model performance. The final performance (using all training data) of the old model where urine output information is not considered is an AUROC of 0.75. The new model with the urine features reaches the same AUROC (0.75) using roughly 21% of the dataset to train on, which equates to multiple years worth of data collection. Hence, the missed benefit is years of having to use the old model until enough data has been gathered for the performance of the new model to surpass that of the old model. We also note that adding the new features in the learning process considerably improves the ultimately achievable prediction performance compared to the old model without urine features: AUROC 0.86 vs. 0.75. Techniques such as transfer learning that can make the training of the new model go faster (e.g., by leveraging the old model) to avoid the missed benefit will clearly help shorten the time to close the gap with the old model.

Figure 2 shows that simple transfer learning enables reaching the same model accuracy with smaller data volumes (i.e., faster, if data is acquired with a new device). The transfer learning model reaches AUROC 0.75 with <18% of the training data as opposed to 21%, which is an improvement, but suggests that there is further benefit to be had from more sophisticated transfer learning approaches.

*Scenario 3: Scarceness of data*. For the AKI model we refer back to the model with urine features in Fig. 1. The results agree with common intuition about relative model performance increasing with the size of datasets, but provide empirical assessment of the magnitude of this effect. For the model with urine features the AUROC varies from 0.66 to 0.86, showing the large impact of available data size. We note that the model does not seem to saturate in terms of possible performance when using the maximum size of data, which suggests that the AKI model is likely to suffer from a lack of data even when trained on huge single datasets like MIMIC-III.

One way of overcoming this lack of data is to develop techniques that can train on datasets from different hospitals, while taking into account the logistical and privacy issues that arise from that. For completeness we also show the performance of the model without urine features as the training dataset size varies. The model accuracy improves less rapidly as the model has less information available and thus cannot generalize as well.

The results for the readmission model are shown in Fig. 3. There are two noticeable differences with the AKI case. Firstly, the increase in performance as data size grows is more rapid when only small proportions of the data are available. Secondly, all of the models more or less saturate at about 40% of the training data, or ~24,000 ICU stays.

This suggests that, despite having a prevalence roughly similar to AKI, readmission can be predicted with a reasonable level of performance by hospitals working independently using this model.

**Table 6 Readmission models: impact of drift and ethnicity.**

| Experiment | Model | | | | | | |
| --- | --- | --- | --- | --- | --- | --- | --- |
| | 24 h | 48 h | 72 h | 24–72 h | 7 days | 30 days | Bounce-back |
| Baseline | 0.71 | 0.74 | 0.76 | 0.76 | 0.77 | 0.75 | 0.84 |
| Drift | 0.68 | 0.71 | 0.74 | 0.73 | 0.75 | 0.74 | 0.81 |
| White-Black | 0.71 | 0.74 | 0.74 | 0.73 | 0.74 | 0.73 | 0.83 |
| White-Latinx | 0.69 | 0.72 | 0.75 | 0.74 | 0.75 | 0.70 | 0.82 |

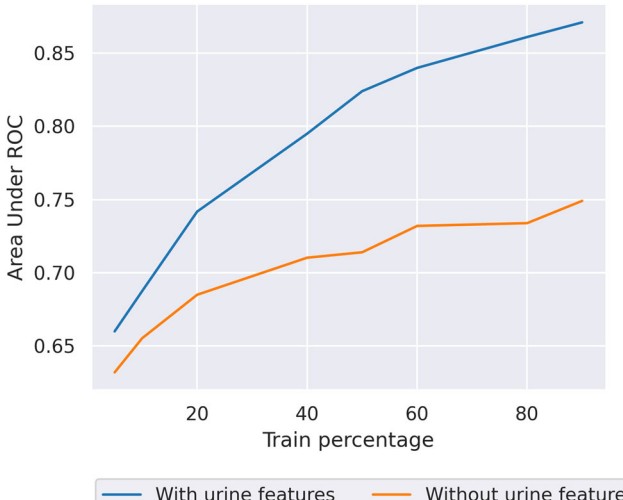

**Fig. 1 AKI model performance as available data size varies, with and without extra features.** Area under receiver operator curve (AUROC) of acute kidney injury (AKI) model performance against dataset size, with (blue line) and without (orange line) urine features. The model with urine features needs to be trained with about 20% of the dataset to achieve the same performance as the model without the urine features trained on 100% of the data, showing that there is a substantial restart cost when adding new features to a model in the absence of transfer learning. Source data is in Supplementary Data 1.

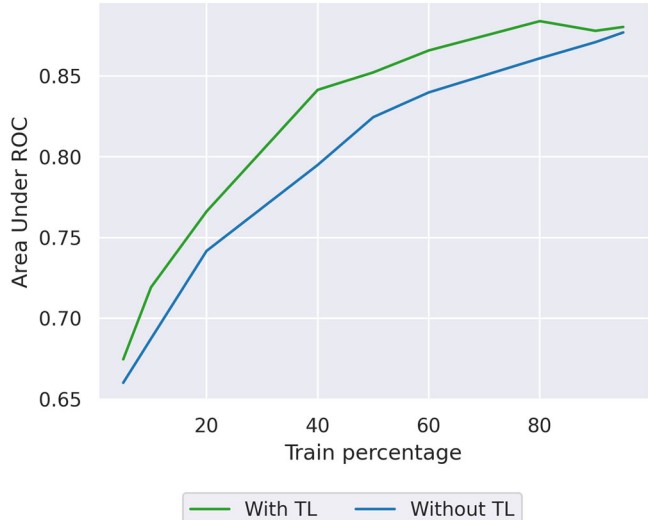

**Fig. 2 AKI model performance as available data size varies, with and without TL.** Area under receiver operator curve (AUROC) of acute kidney injury (AKI) against dataset size, without transfer learning (TL), i.e., the baseline model (blue line), and with simple TL (green line). The transferred model needs less data to achieve the same performance as the model retrained from scratch, showing that there is an advantage in using transfer learning in cases where little data is present. Source data is in Supplementary Data 2.

Though, with about 1000 patients per year in large hospitals, even then retrospective data storage is a must. However, we do not rule out that a more high capacity model could be able to get better performance at the cost of needing more training data, which would make readmission more similar to the AKI use-case with respect to the need for data.

*Scenario 4: Change of population.* The results in Table 5 show the impact in terms of AUROC performance for the AKI model. Although the drop in performance from the demographic split is less dramatic than the effect of drift, the loss of 0.08 and 0.07 AUROC is quite large.

The results for demographic split based on ethnicity for the readmission model are shown in Table 6. The differences between the two alternative models and the baseline is often 0.02 or less, with only 2, resp. 1 entry differing by ≥0.03. Thus, readmission predictability seems to be less affected by this split than the AKI model, and suffers less loss than the split for drift.

## Discussion

In this paper, we explore why AI-driven healthcare is not reaching its full potential in the ICU. Our first step was to speak to stakeholders in the ICU ecosystem (24) to understand the current status and needs, the role of data-driven innovations therein, and why the uptake of these innovations is not larger. Our main conclusions from these interviews are that some AI solutions can be useful in patient care and that blocking factors are mostly technical. Some of these technical blocking factors are more about engineering effort than innovation (e.g., interoperability of data sources within the hospital). Others require innovation at the level of the ML algorithms. These will have to deal with the reality of cross-institutional data silos to mitigate the lack of sufficient data for training, and the discrepancy between clean research data and dirty real-world data. Also ML solutions have to include support for mitigating the impact of different hospital contexts on model reliability. This includes providing tools for generalizing ML solutions to other contexts than those in which they were trained, and drift of datasets over time.

We follow up on several technical points from the interviews by performing ML experiments. We confirm that all four of the issues that we investigated do indeed pose problems for the use of ML in the ICU, and quantify those effects to some extent. Scenario 1, drift, appears to have a large negative impact on model performance. In Scenario 2, adding features can have a large positive impact. The time required to gather enough data to start to benefit from this impact can be large, and there is a clear need for tools to reduce this waiting period. Scenario 3, scarceness of data, shows that a lack of data can be problematic depending on the condition and/or the model. For conditions that are hard but

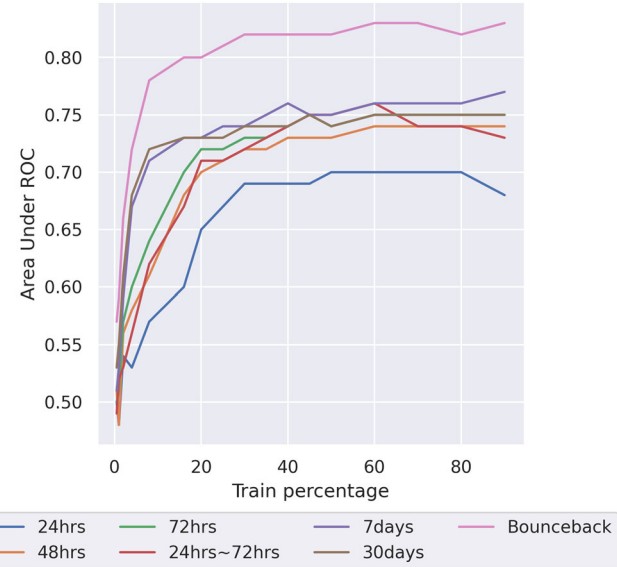

**Fig. 3 Readmission prediction model performance as available data size varies.** Area under receiver operator curve (AUROC) of model performance against dataset size for the readmission prediction models. The models show are readmission after 24 h (blue line), 48 h (orange line), 72 h (green line), from 24 to 72 h (red line), after 7 days (purple line), 30 days (brown line) and bounceback (pink line). Source data is in Supplementary Data 3.

possible to model, a high capacity model (e.g., a deep-neural network) will give the best accuracy but can require a lot of data to train. We conjecture that some models are going to need access to multiple large institutions worth of data to reach their full performance potential, given that >10 years worth of records at a very large hospital (i.e., the MIMIC-III dataset) appears not to be enough to reach saturation for prediction performance on a model for AKI, a medical condition that is only uncommon rather than rare (although admittedly fairly complex). This is also indicated by the VoC results, with insufficient data being a common reason given for the lack of usable ML models. Given the many rules and restrictions around handling of patient data, it is unlikely that it will be possible to pool datasets from different healthcare providers to assemble the large amounts of data needed. Techniques for privacy-preserving learning over multiple parties such as federated learning can be a solution in this case[35]. For other situations with more simple conditions or simpler models, we see it is possible to reach training saturation with data from a single (albeit large) institution. For Scenario 4, we see that a change of population can have a large impact, but also that this appears to have somewhat less impact on simpler models.

We select models to be able to run ML experiments, and to do that we set up criteria for selecting models published in the scientific literature to work with (clinical relevance and reproducibility). We survey available predictive models according to these criteria and find that the current offering is weak. Clearly scientific reproducibility in this area is very limited,. The community could do better in making its research results more accessible. When looking for open-source ML models from the literature, our approach was inspired by Systematic Review. We strongly encourage interested members of this community to develop proper protocols for finding public code repositories and to publish systematic reviews of available open-source models, and believe this would be of great benefit to the community. We note with interest the low numbers of suitable open-source models as it gives a strong indication that the

scope for a hospital to bring in a third-party model from the literature (as opposed to, e.g., a fully commercially developed one) that meets some degree of scientific quality control is pretty limited.

There are various technical solutions to the issues quantified in the experimental section. Techniques such as online learning, concept drift, transfer learning and data augmentation are well developed in the general ML literature. The results of our interviews suggest that there is plenty of scope to use these techniques in the ICU, but for some reason they are not used in practice. Perhaps a larger number of practical experiments can be done in the future to demonstrate how these techniques apply to complex clinical data.

Techniques for learning models across multiple privacy constrained databases are substantially newer, and it would appear that there is plenty more research to be done to further develop these techniques and show how well they apply to ICU data[36]. We look forward with interest to the fruit of that research.

This work presented has some limitations. The VoC study targets a set of 24 stakeholders, which are either ICUs or active in the ICU ecosystem. While on the large side for such a study, 24 is still a small number when it comes to statistics. Moreover, since 13 of these stakeholders are active in Belgium only, there is a possibility of a geographical bias. Expanding the VoC to a more worldwide, or at least European, sample set could inform on the size of this bias. Age was not included in the demographics in the study, although age might be a strong denominator for technical skills and for the general belief of added value. This may bias the responses received.

Some of our experiments are limited by the fact that the dataset was collected from one hospital. In particular, investigating the transfer from one hospital to another by using demographic subsetting of the dataset is a less than ideal proxy for a genuine transfer experiment. Future versions of the this experiment would be better done using the eICU dataset[22,23]. We note however, that there are currently even fewer public models available for this dataset than for MIMIC-III, none of which we considered to be clinically relevant and reproducible at the time of doing the experiments. This limits the scope to work with sufficiently interesting third-party models when evaluating the issues that we raise.

Owing to time limitations, we were only able to work with two models for the experiments. It would be nice to expand this to models for other important ICU conditions such as ARDS[37] or delirium[38].

We concentrate on models with completely public code where the scientific claims can be checked by running it on publicly available data. We have not quantified the extent to which non-public code associated with peer-reviewed publications in this area is actually transferred between hospitals, nor how the receiving hospital would rigorously check the validity in the model in the absence of access to the program and data used to build it. However, the general lack of use of ML models in practice in the ICU would suggest that re-use of non-public models is not a frequent occurrence. Certainly the VoC sample did not give any indications on this being a standard approach.

In this work, we do not directly address the problem of explainability of ML models. Putting objective numerical values on explainability is hard, but clearly this does not prevent explainability from being an important consideration; Kaji et al.[39] show that, especially for challenging and clinically relevant models, explainability can help to highlight problems with the undertaking of building ML models, such as the models resorting to shortcuts that result in them essentially parroting the inputs that they receive from clinical experts. Despite their results, we

take heart from the optimism of the authors that ML models will eventually help in the ICU.

## Data availability
The questionnaire used for the VoC is repeated in Supplementary Note 1, and publicly available on github[40]. The summarized questionnaire results are available in this article. Individual questionnaire results are not available because of confidentiality. Data used in experiments were drawn from the Medical Information Mart for Intensive Care III (MIMIC-III)[20,21]. It is a single-center database containing 53,423 distinct ICU stays for patients aged 16 and over between 2001 and 2012 at Beth Israel Deaconess Medical Center, in Boston, Massachusetts. It is available without cost to researchers who complete the appropriate training for data handling. Figures 1, 2 and 3 have associated source data, which can be found in Supplementary Data 1, 2, and 3. All other data are available from the corresponding author (or other sources, as applicable) on reasonable request.

## Code availability
The code for the AKI model and related experiments is published under the AGPL licence in a public github repository[41]. The complete list of features used can also be retrieved there. The code we used for the experiments on the Readmission model is available from Github[42] under an MIT license (which is the same type of license as the repository that it was forked from). The original repository was downloaded from https://github.com/apakbin94/ICU72hReadmissionMIMICIII.

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

## Acknowledgements
This research received funding from the Flemish regional government (AI Research Program).

## Author contribution
The project on data innovation in ICUs, including the combined qualitative–quantitative approach, was conceptualised by E.D'H. The Voice-of-the-Customer fieldwork was carried out by E.D'H. and T.K.. The numerical experiments were carried out by T.J.A.

and I.C. The paper text was written by T.J.A., E.D'H. and I.C. Overall supervision and consulting was carried out by R.W.

## Competing interests

This work was carried out in the context of an innovation derisking track at a non-profit research institute, with the aim of identifying possible future spin-off activities. The exercise finished and no direct follow-up is currently planned. During the spin-off track, publishing an article clashed with potential generation and protection of IP. When we decided to stop the track, it became possible to publish the knowledge gathered as a scientific article. At the time of submission, there are no competing interests.

## Additional information

**Peer review information** *Communications Medicine* thanks Jakob Wollborn and the other, anonymous, reviewer(s) for their contribution to the peer-review of this work. Peer reviewer reports are available.

