## [Peer Review File · Communications Medicine]

Reviewers' comments:

Reviewer #1 (Remarks to the Author):

In this paper, the authors explored the reasons why machine learning models are not put into practice in the ICU settings. A survey was developed using voice-of-the-customer method to collect blocking factors of the slow uptake of ML models, by interviewing 35 ICU, hospital or MedTech staff. Drift of the model over time, lack of interoperability, scarceness of data and change of population were identified as four main barriers to adoption of ML models. Machine learning experiments in the prediction of acute kidney injury and readmission were performed to validate and quantify those four barriers. This study is potentially meaningful in that it can promote the adoption of machine learning models in the real ICU settings. The study can be significantly improved if the following issues are addressed.

Major:

1. The AKI model was applied to quantify scenario 2,3,4; The readmission model was applied to quantify scenario 1,3,4. Why not apply both models to each of the four scenarios?

2. There are previous ML models developed for AKI and readmission prediction, which should be discussed to give context for the implemented models in the paper. For examples in AKI:

Zimmerman, Lindsay P., et al. "Early prediction of acute kidney injury following ICU admission using a multivariate panel of physiological measurements." *BMC medical informatics and decision making* 19.1 (2019): 1-12.

Li, Yikuan, et al. "Early prediction of acute kidney injury in critical care setting using clinical notes." 2018 IEEE International Conference on Bioinformatics and Biomedicine (BIBM). IEEE, 2018.

Tomašev, Nenad, et al. "A clinically applicable approach to continuous prediction of future acute kidney injury." *Nature* 572.7767 (2019): 116-119.

3. The authors should provide more technical details regarding the feature set of AKI models. What is the cutoff time of collecting those variables? How to deal with missing values? Any other preprocessing steps?

4. When quantifying the effect of one barrier, the authors should adjust for other confounding barriers. E.g. when quantifying the effects of change-of-population, the performance of the model may also be affected by drift.

5. The authors should provide more details about the survey itself, which will be a useful guideline to readers who have similar research interest

6. In the validation of scenario 4: change of population, both insurance type and race/ethnicity are not a good proxy to simulate patient populations across different hospitals. The authors shall consider using the eICU Collaborative Research Database, to quantify the change-of-population.

7. It is recommended to summarize the demographics and responses from interviewees to tables. Enumerating all descriptive statistics in the context is wordy and not intuitive.

8. The figures are not in the same format. Figure 2 has frames and vertical guides, while others don't. 'perc' is used in Figure 4, while 'percentage' is used everywhere else.

Minor:

1. Please pay attention to the direction of the left quotation mark. Quotation marks should always face the quoted material.

Reviewer #2 (Remarks to the Author):

Thank you very much for giving me the opportunity to review a manuscript for publication in your prestigious journal. The presented manuscript covers a very relevant topic: blocking factors of machine learning in the intensive care unit. The authors first conducted a voice of customer (VoC) study with 16 hospitals and 7 Medtech companies participating. The study cohort was diverse in location and job description and had a relatively good gender diversity. The identified blocking factors were then quantified with experiments on the MIMIC database on two use cases: acute kidney injury (AKI) and the prediction of readmission to the ICU.

The method of adding a quantitative scope to qualitative research provides a novel approach. However, when comparing the methodological approaches, the qualitative VoC study is more sound than the quantitative experiments. Streamlining the quantitative results and including a data scientist in the editorial process would benefit this study.

The article is well written and gives a good overview of why there exists a slow uptake of machine learning in the ICU, however a few concerns occurred while reviewing the manuscript, some of major importance.

Major Comments

1. Although the mix of qualitative and quantitative methods is a novelty of this paper, the paper would be strong enough if only the VoC study is kept. This would give the authors the possibility to revise their quantitative experiments for another paper, therefore providing more sound methods and results.

Nonetheless, please find below some suggestions to streamline the results for the quantitative experiments.

2. The VoC study included a lot of information about study demographics. However, age was not included although age might be a strong denominator for technical skills and for the general belief of added value. The authors should add this to the study's limitations.

3. Scenario 1 and scenario 4 – as described on page 10 – seem to be the same or too similar to the reader. However, when they are investigated further in the experiments in section 5, their difference is clearer. The authors should adapt the description on page 10.

4. The quantitative experiments lack information about the data availability. Specifically, section 5.2 (Data availability) lacks information about the chosen cohort e.g. including and excluding factors such as length of stay, age etc.

5. The medical relevance of AKI and readmission prediction is explained. However, it remains unclear how the authors extract the information from the MIMIC database. Again,

the chosen subsets for AKI and readmission prediction need information regarding the chosen cohorts (see comment 3)

6. The authors should include the amount of data available for each of the four scenarios investigated, including the train, validation and test splits.

7. The results section is confusing to read. It should be redrafted and streamlined.

8. "In this scenario we divide the MIMIC-III data in various way to simulate the change of patient demographics that could exist between different hospitals" (section 6.4, p.17). Unclear how the data was split.

Minor Comments

9. The AUC graphics have different designs. The authors should adapt a coherent style.

10. Figure 4 has abbreviated axis descriptors: "avg" and "perc". Please use clearer descriptors.

11. The description of results in Section 5 is not coherent and follows different styles. Sometimes sections have a descriptor, see section 6.4.1 or 6.4.2. The authors should strive to present all results in the same manner.

12. "One respondent mentioned the lack of a "killer app"." (p.9) - what does this mean?

Reviewer #3 (Remarks to the Author):

Thank you for the opportunity to review the article. In their thorough review article, Ashby et al. explore challenges of machine learning implementation in the ICU practice. The authors elegantly integrate an interview of health-care professionals and their perception of ML in ICU care. They provide an analysis of blocking factors for ML uptake in the ICU and also use a ML model to simulate a scenario of deployment of ML.

The article is well written, comprehensive in nature and explores the caveats of ML in the ICU at length. The VoC interview portion is original and innovative. Thank you for the opportunity to review the study.

Att. Reviewers for COMMSMED-21-0554-T
Communications Medicine

Dear Reviewers,

Please find below our point-by-point rebuttal for the article “Machine Learning in the Intensive Care Unit: Survey, blocking factors and quantified needs”. In this article we report on a Voice-of-the-Customer (VoC) study with 40 members of the Intensive Care (IC) ecosystem. The goal was to understand why we see so little analytics solutions in the daily practice of IC units. To understand the blocking factors we identified, we carried out a number of experiments on open ICU data that simulate realistic scenarios derived from the VoC.

In order to improve the quality of our article, we thoroughly reworked it according to the comments of the reviewers and our own critical reading. An overview of the main changes is listed first. This should give the reviewers a rough idea of how the paper has evolved, and be used as the main guideline for identifying differences with the first version. (We tried but gave up on highlighting differences in the text itself, as due to the substantial rewriting we did the result was cluttered rather than enlightening.) Next we repeat the full reviews, adding inline after each point what we did to follow up on the feedback given. This gives each reviewer a detailed view on how their feedback was used to improve the article.

My co-authors and I would like to thank you for the constructive feedback received which has enabled us to improve the article considerably. We look forward to your opinion on this improved version.

Kind regards,

ELLIE D'HONDT
Data innovation catalyst
Exascience Life Lab
T +32 16 28 10 83

ellie.dhondt@imec.be | www.imec.be
Kapeldreef 75 | 3001 Leuven | Belgium Belgium

Main changes.

- The article contains a revised and corrected presentation of the statistics of the VoC study (Sec.3.3-3.4).
- We reworked Sec. 4.1 “Model selection” into a separate and expanded Sec. 5 “Searching for models”. This includes the rationale and criteria for selecting suitable analytics models as a driver for the scenarios extracted from the VoC and for the experiments carried out in later sections (Secs. 5.1 and 5.2). We describe the approach used for this structured search and its results (Sec. 5.3).
- We carried out additional experiments so that the 2 chosen models (AKI and readmission) are run with all scenarios identified in the VoC. One exception is readmission prediction with for the change-of-features scenario, as we could not identify a realistic setup where such a change of features would be relevant. In addition, we added relevant details to the description of experiments s.a. amount of data points, preprocessing steps, etc.
- Added a section on confounding effects between different ways of splitting data (Sec. 8.5).
- Included two appendices: one with the VoC interview questions, and another detailing the most promising models in terms of the structured literature search mentioned earlier.

Full Reviews and point-by-point actions taken.

Reviewer 1 (ML, computational):

Summary. In this paper, the authors explored the reasons why machine learning models are not put into practice in the ICU settings. A survey was developed using voice-of-the-customer method to collect blocking factors of the slow uptake of ML models, by interviewing 35 ICU, hospital or MedTech staff. Drift of the model over time, lack of interoperability, scarceness of data and change of population were identified as four main barriers to adoption of ML models. Machine learning experiments in the prediction of acute kidney injury and readmission were performed to validate and quantify those four barriers. This study is potentially meaningful in that it can promote the adoption of machine learning models in the real ICU settings. The study can be significantly improved if the following issues are addressed.

Major concerns:

1. *The AKI model was applied to quantify scenario 2,3,4; The readmission model was applied to quantify scenario 1,3,4. Why not apply both models to each of the four scenarios?*

We carried out additional experiments so that the 2 chosen models (AKI and readmission) are run in a similar way for all scenarios identified in the VoC. One exception is readmission prediction with for the change-of-features scenario, as we could not identify a realistic setup where such a change of features would be relevant.

2. *There are previous ML models developed for AKI and readmission prediction, which should be*

discussed to give context for the implemented models in the paper. For example in AKI: Zimmerman, Lindsay P., et al. "Early prediction of acute kidney injury following ICU admission using a multivariate panel of physiological measurements." *BMC medical informatics and decision making* 19.1 (2019): 1-12.

Li, Yikuan, et al. "Early prediction of acute kidney injury in critical care setting using clinical notes." 2018 *IEEE International Conference on Bioinformatics and Biomedicine (BIBM)*. IEEE, 2018.

Tomašev, Nenad, et al. "A clinically applicable approach to continuous prediction of future acute kidney injury." *Nature* 572.7767 (2019): 116-119.

These references were included in the article. In addition, the novel and expanded Sec.5 'Searching for models' clarifies why and how we selected the models used in our experiments.

3. The authors should provide more technical details regarding the feature set of AKI models. What is the cutoff time of collecting those variables? How to deal with missing values? Any other preprocessing steps?

We added more details to the description of experiments both for AKI as well as readmission. For the readmission model we refer to the technical setup in the original work for that model.

4. When quantifying the effect of one barrier, the authors should adjust for other confounding barriers. E.g. when quantifying the effects of change-of-population, the performance of the model may also be affected by drift.

The article comments on confounding effects between different ways of splitting data (new section 8.5).

5. The authors should provide more details about the survey itself, which will be a useful guideline to readers who have similar research interest.

The full VoC interview is included in the appendix. Some details and a reference to the appendix were added in the text.

6. In the validation of scenario 4: change of population, both insurance type and race/ethnicity are not a good proxy to simulate patient populations across different hospitals. The authors shall consider using the eICU Collaborative Research Database, to quantify the change-of-population.

We considered eICU as a basis for our experiments for the reasons the reviewer cites. We did this both for the first submission of the paper as well as for this review. The process of this consideration, and the reasons for choosing MIMIC-III, were not well elaborated in the previous version of the paper. This is remediated in this version of the text.

In summary, to go from the results of the VoC study to the experiments, we derived a number of scenarios and typical ways of arriving at models to study these scenarios in. These scenarios pertain to loss of model performance over time or when the population changes. To arrive at the models themselves stakeholders either 1) develop their own or 2) look for models developed elsewhere. So that is what we also did. To make this setup realistic, we derived from the VoC a number of selection criteria which are grouped into what we call *clinical relevance* and *reproducibility*. With these criteria in hand, a structured search was carried out which allowed us to come up with a list of candidate models. We conclude there is a general lack of available models that are reproducible and clinically relevant. This situation requires more discussion and action within the community of people working in this area. It is the main reason why we chose MIMIC-III as an open dataset to base our experiments on, rather than the newer eICU. The notion to re-use existing models in a local hospital context seems very idealistic indeed, with lack of openness and

absent licences being a blocking factor long before suboptimal performance of machine learning models is.

In the text, these results are presented as follows. We reworked Sec. 4.1 “Model selection” into a separate and expanded Sec. 5 “Searching for models”. This includes the rationale and criteria for selecting suitable analytics models as a driver for the scenarios extracted from the VoC and for the experiments carried out in later sections. , criteria which are made concrete in Secs. 5.1 and 5.2. We describe the approach used for this structured search and its results in Sec. 5.3. An appendix completes these sections by detailing the most promising models in terms of the structured search mentioned earlier.

7. It is recommended to summarize the demographics and responses from interviewees to tables. Enumerating all descriptive statistics in the context is wordy and not intuitive.

The article contains a revised, more complete and corrected presentation of the statistics of the VoC study (Sec.3.3-3.4). This involves 3 Tables, showing the statistical characteristics of the set of stakeholders, interviewees and interviews. In terms of completeness, we included 2 more interviews that were not represented in the first version of this article, and we added percentages in all tables.

8. The figures are not in the same format. Figure 2 has frames and vertical guides, while others don't. 'perc' is used in Figure 4, while 'percentage' is used everywhere else.

Figures are now following the same style throughout the paper.

Minor concerns:

Please pay attention to the direction of the left quotation mark. Quotation marks should always face the quoted material.

This has been corrected throughout the text.

Reviewer 2 (Intensive care (clinical), qualitative research on ML in ICU):

Summary. Thank you very much for giving me the opportunity to review a manuscript for publication in your prestigious journal. The presented manuscript covers a very relevant topic: blocking factors of machine learning in the intensive care unit. The authors first conducted a voice of customer (VoC) study with 16 hospitals and 7 Medtech companies participating. The study cohort was diverse in location and job description and had a relatively good gender diversity. The identified blocking factors were then quantified with experiments on the MIMIC database on two use cases: acute kidney injury (AKI) and the prediction of readmission to the ICU.

The method of adding a quantitative scope to qualitative research provides a novel approach. However, when comparing the methodological approaches, the qualitative VoC study is more sound than the quantitative experiments. Streamlining the quantitative results and including a data scientist in the editorial process would benefit this study.

The article is well written and gives a good overview of why there exists a slow uptake of machine learning in the ICU, however a few concerns occurred while reviewing the manuscript, some of major importance.

Major concerns:

1. *Although the mix of qualitative and quantitative methods is a novelty of this paper, the paper would be strong enough if only the VoC study is kept. This would give the authors the possibility to revise their quantitative experiments for another paper, therefore providing more sound methods and results. Nonetheless, please find below some suggestions to streamline the results for the quantitative experiments.*

The editor explicitly advised us '[...] that you strengthen your quantitative data according to the reviewers' suggestions, rather than removing it.'

2. *The VoC study included a lot of information about study demographics. However, age was not included although age might be a strong denominator for technical skills and for the general belief of added value. The authors should add this to the study's limitations.*

The article contains a revised, more complete and corrected presentation of the statistics of the VoC study (Sec.3.3-3.4), including statistical characteristics of stakeholders, interviewees and interviews in tabular form. Indeed we did not include age, which is a limitation. While seniority can be seen as a proxy, this only underlines the strong denominator for skills and vision, as the reviewer suggests. We have added this shortcoming to the limitations in Sec.9.

3. *Scenario 1 and scenario 4 – as described on page 10 – seem to be the same or too similar to the reader. However, when they are investigated further in the experiments in section 5, their difference is clearer. The authors should adapt the description on page 10.*

The text was adapted to clarify the distinction between the two scenarios.

4. *The quantitative experiments lack information about the data availability. Specifically, section 5.2 (Data availability) lacks information about the chosen cohort e.g. including and excluding factors such as length of stay, age etc.*

We added more details to the description of experiments both for AKI as well as readmission. For the readmission model we refer to the technical setup in the original work for that model.

5. *The medical relevance of AKI and readmission prediction is explained. However, it remains unclear how the authors extract the information from the MIMIC database. Again, the chosen subsets for AKI and readmission prediction need information regarding the chosen cohorts (see comment 3).*

See previous comment.

6. *The authors should include the amount of data available for each of the four scenarios investigated, including the train, validation and test splits.*

We use a random split for train and test of 80%, resp. 20% for the baseline performance. We did not consider validation sets for our experiments. For the various data splits, we have added the size of the different splits. In scenario 3 'Scarceness of data' we use percentages and do not state all the numbers, as this would be a bit redundant.

7. *The results section is confusing to read. It should be redrafted and streamlined.*

The results section has been substantially edited. We hope this cleared up any confusion for the reader.

8. "In this scenario we divide the MIMIC-III data in various way to simulate the change of patient demographics that could exist between different hospitals" (section 6.4, p.17). Unclear how the data was split.

This sentence now reads: 'In this scenario we simulate the change of patient demographics that could exist between different hospitals by splitting MIMIC-III data by ethnicity.'

In the model subsections we give further details on which ethnicities were available and how they figure in the experiments.

Minor concerns:

9. The AUC graphics have different designs. The authors should adapt a coherent style.

Figures are now following the same style throughout the paper.

10. Figure 4 has abbreviated axis descriptors: "avg" and "perc". Please use clearer descriptors.

We have changed axis labels to 'Area Under Roc' and 'Train percentage' respectively.

11. The description of results in Section 5 is not coherent and follows different styles. Sometimes sections have a descriptor, see section 6.4.1 or 6.4.2. The authors should strive to present all results in the same manner.

Sections are presented in the same manner.

12. "One respondent mentioned the lack of a "killer app"."(p.9) - what does this mean?

A "killer app" is a marketing term. In this context, one can understand this as an app (a clinical piece of software, e.g. a prediction model) which is so necessary or desirable for clinical care that it is a core driver to innovation and its uptake inevitable. The text was adapted to clarify this.

Reviewer 3 (Critical care (clinical)):

Summary. Thank you for the opportunity to review the article. In their thorough review article, Ashby et al. explore challenges of machine learning implementation in the ICU practice. The authors elegantly integrate an interview of health-care professionals and their perception of ML in ICU care. They provide an analysis of blocking factors for ML uptake in the ICU and also use a ML model to simulate a scenario of deployment of ML.

The article is well written, comprehensive in nature and explores the caveats of ML in the ICU at length. The VoC interview portion is original and innovative. Thank you for the opportunity to review the study.

No remarks to remediate in the text.

REVIEWERS' COMMENTS:

Reviewer #1 (Remarks to the Author):

The authors have addressed all comments from this reviewer.

Reviewer #2 (Remarks to the Author):

Thank you very much for giving me the opportunity to re- review the manuscript. The comments that were given in my first review were addressed accordingly and sufficiently. I do not have further comments at this point.